# Heterologous 1,3-Propanediol Production Using Different Recombinant *Clostridium beijerinckii* DSM 6423 Strains

**DOI:** 10.3390/microorganisms11030784

**Published:** 2023-03-18

**Authors:** Teresa Schoch, Tina Baur, Johanna Kunz, Sophia Stöferle, Peter Dürre

**Affiliations:** Institut für Mikrobiologie und Biotechnologie, Universität Ulm, 89081 Ulm, Germany

**Keywords:** *Clostridium beijerinckii*, *Clostridium pasteurianum*, *Clostridium diolis*, 1,3-propanediol production, glycerol fermentation, promoter effects

## Abstract

1,3-propanediol (1,3-PDO) is a valuable basic chemical, especially in the polymer industry to produce polytrimethylene terephthalate. Unfortunately, the production of 1,3-PDO mainly depends on petroleum products as precursors. Furthermore, the chemical routes have significant disadvantages, such as environmental issues. An alternative is the biobased fermentation of 1,3-PDO from cheap glycerol. *Clostridium beijerinckii* DSM 6423 was originally reported to produce 1,3-PDO. However, this could not be confirmed, and a genome analysis revealed the loss of an essential gene. Thus, 1,3-PDO production was genetically reinstalled. Genes for 1,3-PDO production from *Clostridium pasteurianum* DSM 525 and *Clostridium beijerinckii* DSM 15410 (formerly *Clostridium diolis*) were introduced into *C. beijerinckii* DSM 6423 to enable 1,3-PDO production from glycerol. 1,3-PDO production by recombinant *C. beijerinckii* strains were investigated under different growth conditions. 1,3-PDO production was only observed for *C. beijerinckii* [pMTL83251_P*_pta-ack_*_1,3-PDO.diolis], which harbors the genes of *C. beijerinckii* DSM 15410. By buffering the growth medium, production could be increased by 74%. Furthermore, the effect of four different promoters was analyzed. The use of the constitutive *thlA* promoter from *Clostridium acetobutylicum* led to a 167% increase in 1,3-PDO production compared to the initial recombinant approach.

## 1. Introduction

The demand for 1,3-propanediol (1,3-PDO) has been increasing in the last few years and will continue to rise in the coming years. With a “Compound Annual Growth Rate” of 14.2% as of 2018, an optimistic market expectation for 1,3-PDO is US$ 1,442.77 million in 2027 [1]. 1,3-PDO is an important basic chemical and is widely used as an organic solvent in the food, cosmetics, and pharmaceutical industries. However, the main application of 1,3-PDO is in the polymer industry as a raw material or intermediate, especially as a key monomer in the production of polytrimethylene terephthalate [2,3]. In the past, ethylene oxide hydroformylation and acrolein hydration-hydrogeneration were the two processes mainly used for the chemical synthesis of 1,3-PDO. Both processes depend on petroleum products as precursors for 1,3-PDO production [4,5,6]. However, those processes have many disadvantages, such as high investment, technical difficulties, substrate toxicity, and environmental issues [2,7]. Therefore, a biobased synthesis of 1,3-PDO is desirable.

A biosynthetic 1,3-PDO production approach was developed by DuPont and Genencor in the early 2000s using recombinant *Escherichia coli* cells and glucose as substrate [8]. The process was commercialized by DuPont Tate & Lyle in Loudon (USA) in 2006, with a production capacity of 63,500 tons per year [9,10,11]. Another possibility is the use of glycerol or crude glycerol as a substrate for the microorganisms [12,13]. About 10% crude glycerol is generated as the main by-product from the biodiesel production. With the expanding biodiesel market, the availability of crude glycerol has increased over the last few years. As a result, the price of crude glycerol is dropping [14,15]. Cheap substrates are of particular importance since the cost of substrates accounts for 50–60% of the total production cost of 1,3-PDO [16]. Thus, the use of cheap glycerol from biodiesel as a substrate for the production of 1,3-PDO could be a good alternative to the original chemical production routes. Therefore, the establishment of a biological production of 1,3-PDO via fermentation of glycerol is desired for a sustainable 1,3-PDO production.

Several microorganisms are known to naturally reduce glycerol to 1,3-PDO including *Klebsiella pneumoniae*, *Citrobacter freundii*, and *Enterobacter agglomerans* [17,18,19]. A big disadvantage of some of these microorganisms is their pathogenicity and the associated restrictions on industrial use [7,19,20]. However, a few clostridial species are also able to use glycerol for the production of 1,3-PDO such as *Clostridium pasteurianum*, *Clostridium butyricum*, *Clostridium diolis* (reclassified now as *Clostridium beijerinckii* [21]), and some *Clostridium beijerinckii* strains [22,23,24,25,26]. These different clostridial species harbor different glycerol reduction pathways. In the case of *C. pasteurianum*, glycerol is dehydrated via a coenzyme B_12_-dependent glycerol dehydratase (DhaB, DhaC, DhaE) to 3-hydroxypropionaldehyde (3-HPA), followed by the conversion of 3-HPA via a NADH-linked 1,3-PDO dehydrogenase (DhaT) to 1,3-PDO [27]. The glycerol reductive pathway of *C. butyricum* and *C. beijerinckii* consists of a coenzyme B_12_-independent glycerol dehydratase (DhaB1, DhaB2) for the dehydration of glycerol to 3-HPA, also followed by the conversion of 3-HPA to 1,3-PDO via a NADH-linked 1,3-PDO dehydrogenase (DhaT) [28,29]. However, glycerol is not only reduced to 1,3-PDO but also oxidized and used for biomass and by products e.g., acetate, butyrate, ethanol, and butanol [30]. An overview of the glycerol consumption pathway is given in Figure 1.

In the past, most publications focused on the conversion of glycerol to 1,3-PDO with wild-type clostridial strains. Clostridia are difficult to manipulate genetically, due to the limited availability of genetic tools as well as the occurrence of native restriction-modification systems, resulting in major obstacles in obtaining recombinant strains [31,32]. Therefore, development of new genetic tools and methods to overcome restriction-modification systems are essential for the construction of an optimized production strain for the waste valorization from glycerol to 1,3-PDO in the future. One of these new developments is the newly published transformation protocol for *C. beijerinckii* DSM 6423 [31], which was used in this study to create recombinant 1,3-PDO production strains.

The two different 1,3-PDO production pathways form *C. pasteurianum* DSM 525 and *C. beijerinckii* DSM 15410 were expressed in *C. beijerinckii* DSM 6423 in a plasmid-based manner. Furthermore, the effect of different promoters on 1,3-PDO production was examined in this study. The results presented here demonstrate the heterologous production of 1,3-PDO with genetically engineered *C. beijerinckii* DSM 6423 strains for the first time.

## 2. Material & Methods

### 2.1. Strains and Cultivation

Bacterial strains used in this study are listed in Table 1. *C. beijerinckii* strains were cultivated under anaerobic conditions at 37 °C. YTG medium was used as a complex medium, for solid medium 2YTG medium containing 1.5% (*w*/*v*) agar was prepared. 2YTG medium contained (per L): tryptone, 16 g; yeast extract, 10 g; NaCl, 5 g; glucose, 20 g [31]. Growth experiments were performed in minimal medium referred to as glycerol medium. Glycerol medium contained (per L): K_2_HPO_4_, 4 g; KH_2_PO_4_, 1.5 g; (NH_4_)_2_SO_4_, 2 g; MgSO_4_ · 7H_2_O, 0.3 g; yeast extract, 1 g; cysteine-HCl · H_2_O, 1.2 g; resazurin, 1 mg; trace element solution, 1 mL. Composition of trace element solution (per L): CoCl_2_ · 2H_2_O, 0.2 g; CuCl_2_ · 2H_2_O, 0.02 g; H_2_BO_3_, 0.06 g; HCl (37%), 0.9 mL; MnCl_2_ · 4H_2_O, 0.1 g; Na_2_MoO_4_ · 2H_2_O, 0.035 g; NiCl_2_ · 6H_2_O, 0.025 g; ZnCl_2_, 0.07 g. In case of growth experiments with buffered medium, glycerol medium was prepared with 10.46 g/L 3-(N-morpholino)propanesulfonic acid (MOPS). For the construction and growth of recombinant *C. beijerinckii* strains, media were supplemented with the appropriate antibiotic. *E. coli* strains were cultivated aerobically under shaking conditions (180 rpm) at 37 °C in LB medium [33]. The solid medium contained 1.5% (*w*/*v*) agar. For cloning purposes, media were supplemented with the respective antibiotic.

### 2.2. Construction of Recombinant C. beijerinckii Strains

Standard molecular cloning techniques were performed according to established protocols [33]. All primers used for the construction of the different 1,3-PDO production plasmids are listed in Table 2. Primers were designed using the “NEBuilder^®^” online tool and synthesized by Biomers.net GmbH (Ulm, Germany). Genomic DNA or plasmid DNA served as a template for amplification via PCR. Amplification was performed using “CloneAmp^TM^ HiFi polymerase” (Takara Bio USA, Inc., Mountain View, CA, USA) or “Phusion^TM^ Green High-Fidelity DNA Polymerase” (Thermo Fischer Scientific Inc., Waltham, MA, USA). Linearization of vectors and plasmids was achieved using “FastDigest^TM^ restriction enzymes” (Thermo Fischer Scientific Inc., Waltham, MA, USA). Purification of DNA fragments from agarose gels or solutions after PCR as well as after plasmid linearization were performed using the “Zymoclean^TM^ Gel DNA Recovery Kit” (ZYMO Research Corp., Irvine, CA, USA) or the “DNA Clean &Concentrator^®^ Kit” (ZYMO Research Corp., Irvine, CA, USA), respectively. The procedure was carried out as described by the manufacturer. Assembly of purified DNA fragments was performed with the “NEBulder^®^ HiFi DNA Assembly Master Mix” (New England Biolabs^®^ Inc., Ipswich, MA, USA), according to the manufacturer’s instructions. After assembly, 3.5-5 µL of cloning mixture were used to transform the chemically competent *E. coli* strain XL1-Blue MRF’. Plasmid preparation from *E. coli* cells was performed using the “Zyppy^TM^ Plasmid Miniprep Kit” (ZYMO Research Corp., Irvine, CA, USA) following the manufacturer´s instructions. The accuracy of the desired plasmid was checked by an analytic digestion, again using “FastDigest^TM^ restriction enzymes” (Thermo Fischer Scientific Inc., Waltham, MA, USA). Subsequently, plasmids that showed the expected DNA fragments were sent to Microsynth AG (Balgrach, Switzerland) for sequencing. An overview of the newly constructed plasmids and their relevant characteristics is shown in Table 1.

To construct pMTL83251_P*_pta-ack_*_1,3-PDO.diolis, the plasmid pMTL83251_P*_pta-ack_*_1,3-PDO.CLOBI was digested using the restriction enzymes *Xba*I, *Hind*III, and *Nhe*I to remove the 1,3-PDO gene cluster from *C. beijerinckii* DSM 6423. The PCR-amplified 1,3-PDO gene cluster from *C. beijerinckii* DSM 15410 (locus tag K684DRAFT_00976-00979; primers: dhaB1/2CoT.diol_fwd and dhaB1/2CoT.diol_rev) was cloned into the digested pMTL83251_P*_pta-ack_*_1,3-PDO.CLOBI, still harboring the *pta-ack* promoter from *C. ljungdahlii*. For cloning of pMTL83251_P*_thlA_*_dhaBCET.Cpas, plasmid pMTL83251 was linearized using restriction enzymes *Sal*I and *Sma*I. Digested plasmid was ligated with P*_thlA_* (primers: PthlA.Cpas_fwd and PthlA.Cpas_rev) amplified from *C. acetobutylicum* as well as *dhaBCET* (locus tags: Ga0078015_112319-112317; primers: dhaBCE_fwd and dhaBCE_rev) and *dhaT* (locus tag: Ga0078015_112312; primers: dhaT_fwd and dhaT_rev) DNA fragments amplified from genomic DNA from *C. pasteurianum* DSM 525. Plasmids for testing the effects of different promoter sequences were based on the plasmid pMTL83251_P*_pta-ack_*_1,3-PDO.diolis. The *pta-ack* promoter was removed using the restriction enzymes *Sma*I and *Xba*I. The P*_thlA_* fragment from *C. acetobutylicum* was amplified from the plasmid pMTL83251_P*_thlA_*_FAST (primers: PthlA_1,3-PDO_fwd and PthlA_1,3-PDO_rev) and ligated into the digested pMTL83251_P*_pta-ack_*_1,3-PDO.diolis, resulting in the plasmid pMTL83251_P*_thlA_*_1,3-PDO.diolis. To construct the plasmid pMTL83251_P*_bld_*_1,3-PDO.diolis, the PCR-amplified P*_bld_* DNA fragment from *C. saccharoperbutylacetonicum* (amplified from pMTL83251_P*_bld_*_FAST; primers: Pbld_1,3-PDO_fwd and Pbld_1,3-PDO_rev) was cloned into the digested pMTL83251_P*_pta-ack_*_1,3-PDO.diolis plasmid. P*_bgal_* form *C. perfringens* was amplified from the plasmid pMTL83251_P*_bgaL_*_FAST using primers PbgaL_1,3-PDO_fwd and PbgaL_1,3-PDO_rev, and ligated into the *Sma*I and *Xba*I digested pMTL83251_P*_pta-ack_*_1,3-PDO.diolis plasmid to assemble the plasmid pMTL83251_P*_bgaL_*_1,3-PDO.diolis.

Prior to the transformation of *C. beijerinckii*, the newly constructed plasmids were transformed into electrocompetent *E. coli* SCS110 cells. Those *E. coli* cells are *dcm* and *dam* deficient, leading to unmethylated plasmid DNA after replication and isolation of plasmids. Transformation of *C. beijerinckii* was performed as described by Diallo et al., 2020 [31].

### 2.3. Strain Verification

Newly constructed *C. beijerinckii* strains and strains used for growth experiments were verified by 16S rRNA gene sequencing, and plasmids were retransformed in *E. coli* XL1-Blue MRF’ for further analysis. Therefore, genomic DNA was isolated using the “MasterPure^TM^ Gram Positive DNA Purification Kit” (Lucigen Corp., Middleton, WI, USA). Genomic DNA served as a template for amplification of the 16S rRNA gene (primers: 16S-27F and 1492r) using the “ReproFast proofreading polymerase” (Genaxxon Bioscience GmbH, Ulm, Germany). The amplified 16S rRNA gene was sequenced by Microsynth AG (Balgrach, Switzerland), and the sequence was blasted using NCBI blastn with the RNA/ITS database. For plasmid verification, *E. coli* XL1-Blue MRF´ was transformed with 3 µL of genomic DNA. After growth, plasmids were isolated and checked via analytic digestion.

### 2.4. Growth Conditions of Batch Experiments

For batch growth in bottles, the different *C. beijerinckii* strains were inoculated in a glycerol medium. The medium was prepared as described above. The pH of the glycerol medium was adjusted to 7.5 with KOH. After preparing the medium, 50 mL aliquots were filled in bottles and closed airtight. The gas phase was exchanged with N_2_:CO_2_ (80:20), and bottles were autoclaved. Before inoculation, the medium was supplemented with 40 mM xylose and 100 mM glycerol. For recombinant *C. beijerinckii* strains, clarithromycin (5 µg/mL) was added to the medium. To analyze the influence of vitamin B_12_ supplementation, 5 mg/L sterile vitamin B_12_ was added to the autoclaved glycerol medium of *C. beijerinckii* [pMTL83251_P*_thlA_*_dhaBCET.Cpas]. OD_600_ and pH were monitored during the growth experiments. 2-mL samples for the analysis of substrate consumption and product concentration were taken. The samples were withdrawn with syringes and frozen until analysis took place.

### 2.5. Analytical Methods

The 2-mL samples withdrawn during growth experiments were thawed and subsequently centrifuged (18,000× *g*; 30 min; 4 °C). Acetate, butyrate, butanol, ethanol, acetoin, and isopropanol concentrations were determined using a “Clarus 600 gas chromatograph” (Perkin Elmer Inc., Waltham, MA, USA) equipped with a flame ionization detector heated to 300 °C and a flowrate of synthetic air of 450 mL min^–1^. H_2_ was used as the carrier gas (45 mL min^–1^). Prior to analysis, 480 μL supernatant was acidified with 20 µL 2 M HCl. 1 µL of acidified supernatant was injected onto an “Elite-FFAP” column (30 m × 0.32 mm; Perkin Elmer Inc., Waltham, MA, USA) with the injector heated to 225 °C. For analysis, the following temperature profile was used: 40 °C for 3 min, 40 °C to 250 °C by 40 °C min^–1^, 250 °C for 1 min. Xylose and 1,3-PDO concentrations were measured using an “Agilent 1260 Infinity Series HPLC” system (Agilent Technologies, Santa Clara, CA, USA) equipped with a refractive index detector and a diode array detector. 20 µL of supernatant were injected into a “CS organic acid” precolumn (40 × 8 mm) followed by a “CS organic acid” column (300 × 8 mm; CS-Chromatographie Service GmbH, Langerwehe, Germany). The column was heated to 40 °C, and a mobile phase consisting of 5 mM H_2_SO_4_ with a flow rate of 0.6 mL min^–1^ was used. Glycerol concentration was determined using the “Glycerol Assay Kit MAK117” (Sigma-Aldrich^®^, St. Louis, MO, USA) following the manufacturer’s manual. For glycerol standards, 1 mM, 0.8 mM, 0.6 mM, 0.4 mM, 0.2 mM, and 0.1 mM were used. Prior to analysis, the supernatant of the growth experiment samples was diluted (1:200 or 1:100) to fit the standard range used for the assay.

## 3. Results

### 3.1. Construction of Recombinant 1,3-PDO Production C. beijerinckii Strains

Two types of glycerol dehydratases are present in different clostridial strains. One is a coenzyme B_12_-dependent glycerol dehydratase, and the other is a coenzyme B_12_-independent glycerol dehydratase. The reported production of 1,3-PDO by *C. beijerinckii* DSM 6423 [37] could not be reproduced. Genome analysis showed that the essential gene encoding 1,3-PDO dehydrogenase was missing (Figure 2A). Therefore, we attempted to reconstitute the 1,3-PDO production ability. To examine whether *C. beijerinckii* DSM 6423 can use both glycerol dehydratases to produce 1,3-PDO, two different plasmids were constructed and transformed into *C. beijerinckii*. The strain *C. beijerinckii* [pMTL83251_P*_thlA_*_dhaBCET.Cpas] contains a plasmid harboring the *dhaB*, *dhaC*, and *dhaE* genes (locus tags: Ga0078015_112319-112317) encoding the coenzyme B_12_-dependent glycerol dehydratase, as well as the *dhaT* gene (locus tag: Ga0078015_112312) encoding the 1,3-PDO dehydrogenase of *C. pasteurianum* DSM 525 [27,38]. The constitutively active *thlA* promoter of *C. acetobutylicum* was used to ensure expression. Figure 2B shows the 1,3-PDO gene cluster of *C. pasteurianum*. The 1,3-PDO gene cluster contains four more genes besides *dhaB*, *dhaC*, *dhaE*, and *dhaT*, which are not present on the plasmid. It is assumed that enzymes encoded by these genes are involved in the reactivation of coenzyme B_12_, due to the high similarity to genes also present in the 1,3-PDO gene cluster of *C. freundii* [39]. In contrast, the second recombinant strain of *C. beijerinckii* [pMTL83251_P*_pta-ack_*_1,3-PDO.diolis] carries the plasmid pMTL83251_P*_pta-ack_*_1,3-PDO.diolis harboring the whole 1,3-PDO gene cluster of *C. beijerinckii* DSM 15410 (formerly *C. diolis* DSM 15410). The 1,3-PDO gene cluster of *C. beijerinckii* DSM 15410 is shown in Figure 2C. The two genes *dhaB1* and *dhaB2* (locus tag: K684DRAFT_00976 and K684DRAFT_00977) encode a coenzyme B_12_-independent glycerol dehydratase, and *dhaT* (locus tag: K684DRAFT_00979) the 1,3-PDO dehydrogenase [28]. Furthermore, the 1,3-PDO gene cluster also includes the *pduO* gene (locus tag: K684DRAFT_00978). This gene is annotated as an ATP:cob(I)alamin adenosyltransferase, necessary for the conversion of vitamin B_12_ to coenzyme B_12_ [40] and is also present on the 1,3-PDO production plasmid. The 1,3-PDO gene cluster was cloned under the control of the constitutively active *pta-ack* promoter of *C. ljungdahlii* [41]. To examine the effect of different promoters on 1,3-PDO production, three additional plasmids were constructed. The plasmid pMTL83251_P*_thlA_*_1,3-PDO.diolis harbors the early growth phase-associated *thlA* promoter of *C. acetobutylicum* [42] instead of the *pta-ack* promoter, likewise pMTL83251_P*_thlA_*_dhaBCET.Cpas. For the plasmid pMTL38251_P*_bld_*_1,3-PDO.diolis, the exponential growth phase-associated *bld* promoter of *C. saccharoperbutylacetonicum* [43] was used. Plasmid pMTL83251_P*_bgaL_*_1,3-PDO.diolis includes the lactose-inducible *bgaL* promoter of *C. perfringens* [44].

### 3.2. Recombinant 1,3-PDO Production

Different batch growth experiments were carried out to evaluate if the constructed recombinant *C. beijerinckii* strains produced 1,3-PDO and under what conditions the highest 1,3-PDO concentration could be achieved. Since *C. beijerinckii* DSM 6423 could not grow with glycerol as the sole carbon source, xylose was also added to the glycerol medium. The first batch growth experiment was performed with unbuffered glycerol medium for cultivation of *C. beijerinckii* [pMTL83251_P*_pta-ack_*_1,3-PDO.diolis], *C. beijerinckii* [pMTL83251], and the wild-type *C. beijerinckii* (Figure 3). Compared to the control strains *C. beijerinckii* [pMTL83251] and the *C. beijerinckii* wild-type, the recombinant strain *C. beijerinckii* [pMTL83251_P*_pta-ack_*_1,3-PDO.diolis] grew faster and reached the stationary phase after 24 h with an OD_600_ of 0.73. However, the three analyzed *C. beijerinckii* strains reached a final OD_600_ of about 1 (Figure 3A). The drastic pH drop during the first 24 h of cultivation was only detected in *C. beijerinckii* [pMTL83251_P*_pta-ack_*_1,3-PDO.diolis] (Figure 3B). Unexpectedly, 1,3-PDO could only be detected in the growth medium of the recombinant strain *C. beijerinckii* [pMTL83251_P*_pta-ack_*_1,3-PDO.diolis]. After almost 250 h of incubation, 18.9 mM 1,3-PDO was measured in case of *C. beijerinckii* [pMTL83251_P*_pta-ack_*_1,3-PDO.diolis] (Figure 3C). The glycerol concentration in the case of *C. beijerinckii* [pMTL83251_P*_pta-ack_*_1,3-PDO.diolis] decreased during the experiment by 33.3 mM. In contrast, the glycerol concentration of the wild-type strain and *C. beijerinckii* [pMTL83251] decreased by 7 mM and 13 mM, respectively (Figure 3D). Xylose was not consumed completely by any of the strains tested. The main product of the analyzed *C. beijerinckii* strains was butyrate (wild-type: 20.9 mM; *C. beijerinckii* [pMTL83251]: 21.3 mM; *C. beijerinckii* [pMTL83251_P*_pta-ack_*_1,3-PDO.diolis]: 20.4 mM) (Figure 3E). Only traces of ethanol were detected for all tested *C. beijerinckii* strains (0.2 mM). The lowest acetate concentration (2.2 mM) was measured in case of the wild-type strain. However, this strain also produced the most butanol, with 4.5 mM during the growth experiment. *C. beijerinckii* [pMTL83251] produced 3.6 mM acetate and 2.9 mM butanol throughout the cultivation. The highest acetate concentration was measured in the growth medium of *C. beijerinckii* [pMTL83251_P*_pta-ack_*_1,3-PDO.diolis] (6.4 mM). However, this strain produced only traces of butanol (0.2 mM).

### 3.3. Influence of Buffered Glycerol Medium on 1,3-PDO Production

Due to the drastic drop in pH values in the first growth experiment (Figure 3B), the effect of MOPS-buffered medium on the production of 1,3-PDO was examined. The following batch growth experiment was executed with the two recombinant strains *C. beijerinckii* [pMTL83251_P*_thlA_*_dhaBCET.Cpas] and *C. beijerinckii* [pMTL83251_P*_pta-ack_*_1,3-PDO.diolis]. As controls, the wild-type strain as well as *C. beijerinckii* [pMTL83251] were used. The results of this batch growth experiment are shown in Figure 4. Again, *C. beijerinckii* [pMTL83251_P*_pta-ack_*_1,3-PDO.diolis] reached the stationary phase after 24 h of incubation (OD_600_: 0.85). In contrast, the OD_600_ of *C. beijerinckii* [pMTL83251_P*_thlA_*_dhaBCET.Cpas] as well as the wild-type strain and *C. beijerinckii* [pMTL83251] decreased after 24 h and 48 h, respectively. Afterwards, the OD_600_ of all three strains rose again. All tested strains reached a similar final OD_600_ after 243 h (Figure 4A). Compared to the first growth experiment, the pH dropped in the case of *C. beijerinckii* [pMTL83251_P*_pta-ack_*_1,3-PDO.diolis] during the first 48 h of incubation only to a pH value of 5.47. Afterwards, the pH value decreased to a final value of 5.39 (Figure 4B). Again, production of 1,3-PDO was only detected in the case of *C. beijerinckii* [pMTL83251_P*_pta-ack_*_1,3-PDO.diolis]. The amount of 1,3-PDO was increased to 32.8 mM 1,3-PDO for the recombinant *C. beijerinckii* strain harbouring the 1,3-PDO production genes of *C. beijerinckii* DSM15410 (Figure 4C). Minor concentrations of 1,3-PDO production could be measured for the wild-type strain, *C. beijerinckii* [pMTL83251], and *C. beijerinckii* [pMTL83251_P*_thlA_*_dhaBCET.Cpas] (0.7, 0.4, and 0.2 mM, respectively). A distinct decrease of 40.1 mM glycerol was only observed in the case of *C. beijerinckii* [pMTL83251_P*_pta-ack_*_1,3-PDO.diolis]. The other three examined *C. beijerinckii* strains *C. beijerinckii* [pMTL83251_P*_thlA_*_dhaBCET.Cpas], *C. beijerinckii* [pMTL83251], and the wild-type strain consumed only minor amounts of glycerol (3.5 mM, 14.1 mM, and 12.6 mM, respectively) (Figure 4D). The supplemented xylose was consumed completely by the strains *C. beijerinckii* [pMTL83251_P*_thlA_*_dhaBCET.Cpas], *C. beijerinckii* [pMTL83251], and the wild-type strain during the growth experiment. In contrast, *C. beijerinckii* [pMTL83251_P*_pta-ack_*_1,3-PDO.diolis] only consumed 18.1 mM of the added xylose (Figure 4D). As in the first growth experiment, butyrate was the main product. The highest butyrate production was detected in the culture broth of *C. beijerinckii* [pMTL83251] with 40.1 mM, followed by *C. beijerinckii* [pMTL83251_P*_pta-ack_*_1,3-PDO.diolis] and *C. beijerinckii* [pMTL83251_P*_thlA_*_dhaBCET.Cpas], which produced 39.6 mM and 38.1 mM butyrate, respectively. The wild-type strain produced 33.3 mM butyrate. The highest acetate production was also detected in the culture broth of the strain *C. beijerinckii* [pMTL83251] at 3.1 mM. The strains *C. beijerinckii* [pMTL83251_P*_pta-ack__*1,3-PDO.diolis], *C. beijerinckii* [pMTL83251_P*_thlA_*_dhaBCET.Cpas], as well as the wild-type strain, produced 2.2 mM, 3.0 mM, and 1,3 mM acetate. Similar to the first growth experiment, the highest butanol concentration was obtained by the wild-type strain (4.1 mM butanol). In the case of *C. beijerinckii* [pMTL83251] and *C. beijerinckii* [pMTL83251_P*_pta-ack_*_1,3-PDO.diolis], 1.5 mM and 0.5 mM butanol were produced. No butanol could be detected in the culture medium of *C. beijerinckii* [pMTL83251_P*_thlA_*_dhaBCET.Cpas]. Only traces of ethanol were detected by all tested *C. beijerinckii* strains (0.4 mM). An overview of the synthesized products is shown in Figure 4E. The use of MOPS-buffered glycerol medium resulted in a 74% increase in 1,3-PDO production using *C. beijerinckii* [pMTL83251_P*_pta-ack_*_1,3-PDO.diolis].

### 3.4. Effect of Vitamin B_12_ Supplementation

Since the used glycerol dehydratase of *C. pasteurianum* is coenzyme B_12_-dependent, the effect of supplementation of vitamin B_12_ in the MOPS-buffered growth medium of *C. beijerinckii* [pMTL83251_P*_thlA_*_dhaBCET.Cpas] was studied. Therefore, the glycerol medium of *C. beijerinckii* [pMTL83251_P*_thlA_*_dhaBCET.Cpas] was supplemented with vitamin B_12_. The results are shown in Figure 5. During this growth experiment, the recombinant strain *C. beijerinckii* [pMTL83251_P*_pta-ack_*_1,3-PDO.diolis] grew slower than during previously described growth experiments. Cells reached the stationary phase after 48 h with an OD_600_ of 0.62. As reported before, a decrease of the OD_600_ after 24 h and 48 h was observed for *C. beijerinckii* [pMTL83251_P*_thlA_*_dhaBCET.Cpas] as well as the wild-type strain and *C. beijerinckii* [pMTL83251]. After 243 h of incubation, the OD_600_ values of *C. beijerinckii* [pMTL83251_P*_thlA_*_dhaBCET.Cpas], *C. beijerinckii* [pMTL83251_P*_pta-ack_*_1,3-PDO.diolis], and the wild-type were similar; only the OD_600_ of *C. beijerinckii* [pMTL83251] was lower with a value of 0.69 (Figure 5A). In parallel to the slower growth, the pH of the culture broth of *C. beijerinckii* [pMTL83251_P*_pta-ack_*_1,3-PDO.diolis] also dropped slower than described for the previous experiments (Figure 5B). Throughout the growth experiment, *C. beijerinckii* [pMTL83251_P*_pta-ack_*_1,3-PDO.diolis] produced 43.1 mM 1,3-PDO. Again, the three other *C. beijerinckii* strains synthesized only traces of 1,3-PDO (wild-type and *C. beijerinckii* [pMTL83251]: 0.7 mM; *C. beijerinckii* [pMTL83251_P*_thlA_*_dhaBCET.Cpas]: 0.6 mM) (Figure 5C). The 1,3-PDO-producing strain *C. beijerinckii* [pMTL83251_P*_pta-ack_*_1,3-PDO.diolis] consumed 66.9 mM glycerol during batch growth. *C. beijerinckii* [pMTL83251_P*_thlA_*_dhaBCET.Cpas] used 27.3 mM glycerol. The two control strains consumed only around 10 mM glycerol. In contrast, xylose was consumed completely by the wild-type strain, *C. beijerinckii* [pMTL83251], and *C. beijerinckii* [pMTL83251_P*_thlA_*_dhaBCET.Cpas] (Figure 5D). In the case of *C. beijerinckii* [pMTL83251_P*_pta-ack_*_1,3-PDO.diolis], 1,3-PDO was the main product. Compared to the other three tested strains, *C. beijerinckii* [pMTL83251_P*_pta-ack_*_1,3-PDO.diolis] showed the lowest butyrate (19.9 mM) and acetate (1.1 mM) concentrations. The highest butyrate and acetate amounts were observed with the control strain *C. beijerinckii* [pMTL83251], i.e., 23.6 mM butyrate and 2.4 mM acetate. *C. beijerinckii* [pMTL83251_P*_thlA_*_dhaBCET.Cpas] produced 20.9 mM butyrate and 1.6 mM acetate during growth. The wild-type strain produced butyrate (22.7 mM), acetate (1.5 mM) and butanol (1.4 mM). Traces of butanol were also observed in cultures of *C. beijerinckii* [pMTL83251_P*_pta-ack_*_1,3-PDO.diolis] (0.4 mM). All examined *C. beijerinckii* strains synthesized traces of ethanol (0.3–0.2 mM). An overview of the products is provided in Figure 5E. Overall, a benefit of the vitamin B_12_ supplementation could not be observed.

### 3.5. Effect of Different Promoters on Recombinant 1,3-PDO Production

To exclude that the *thlA* promoter is not functional in *C. beijerinckii* and therefore *C. beijerinckii* [pMTL83251_P*_thlA_*_dhaBCET.Cpas] could not produce 1,3-PDO in the previously performed batch growth experiments, and to examine if other promoters have a positive effect on the 1,3-PDO production, four different promoters were tested. The batch experiments were again performed with MOPS-buffered medium using the recombinant strains *C. beijerinckii* [pMTL83251_P*_pta-ack_*_1,3-PDO.diolis], *C. beijerinckii* [pMTL83251_P*_bgaL_*_1,3-PDO.diolis], *C. beijerinckii* [pMTL83251_P*_bld_*_1,3-PDO.diolis], *C. beijerinckii* [pMTL83251_P*_thlA_*_1,3-PDO.diolis], as well as the wild-type *C. beijerinckii* strain (Figure 6). *C. beijerinckii* [pMTL83251_P*_thlA_*_1,3-PDO.diolis] reached the stationary phase after 10 h, whereas *C. beijerinckii* [pMTL83251_P*_pta-ack_*_1,3-PDO.diolis] reached the stationary phase after 24 h. 1,3-PDO gene expression of *C. beijerinckii* [pMTL83251_P*_bgaL_*_1,3-PDO.diolis] was induced via supplementation of 20 mM lactose after 10 h of cultivation (OD_600_: 0.19). Before induction, the *C. beijerinckii* [pMTL83251_P*_bgaL_*_1,3-PDO.diolis] cultures grew similarly. However, in contrast to the induced cultures, the OD_600_ of the uninduced strain did not increase but even decreased. After 48 h, the OD_600_ of the uninduced strain *C. beijerinckii* [pMTL83251_P*_bgaL_*_1,3-PDO.diolis] increased again. The strains containing the inducible *bgaL* promoter showed the lowest final OD_600_ (induced: 0.61; uninduced: 0.57) (Figure 6A). As observed in the growth experiments before, the pH of the 1,3-PDO production strains dropped faster than the pH values of the control cultures (Figure 6B). 1,3-PDO was produced by all recombinant production strains. Surprisingly, the 1,3-PDO production strain harbouring the *thlA* promoter and the 1,3-PDO genes of *C. beijerinckii* DSM 15410 produced at 50.4 mM the highest 1,3-PDO concentration after 240 h of incubation. Throughout the course of the experiment, *C. beijerinckii* [pMTL83251_P*_bld_*_1,3-PDO.diolis] produced 37.4 mM 1,3-PDO. 36.3 mM 1,3-PDO were detected in induced cultures of *C. beijerinckii* [pMTL83251_P*_bgaL_*_1,3-PDO.diolis]. In contrast, the uninduced strain *C. beijerinckii* [pMTL83251_P*_bgaL_*_1,3-PDO.diolis] only produced 2.1 mM 1,3-PDO. During this growth experiment, *C. beijerinckii* [pMTL83251_P*_pta-ack_*_1,3-PDO.diolis] produced only 27.3 mM 1,3-PDO. In the wild-type, traces of 1,3-PDO could be detected (0.3 mM). The 1,3-PDO values produced by the different tested strains are shown in Figure 6C. Throughout the growth experiment, *C. beijerinckii* [pMTL83251_P*_thlA_*_1,3-PDO.diolis] consumed 63.4 mM glycerol (Figure 6D). The induced *C. beijerinckii* [pMTL83251_P*_bgaL_*_1,3-PDO.diolis] as well as *C. beijerinckii* [pMTL83251_P*_bld_*_1,3-PDO.diolis] showed similar glycerol consumption (53.2 mM and 52.1 mM, respectively). Surprisingly, the glycerol concentration in the culture broth of *C. beijerinckii* [pMTL83251_P*_pta-ack_*_1,3-PDO.diolis] decreased by only 37.6 mM, matching the low 1,3-PDO production during growth (Figure 6D). Again, the non-1,3-PDO-producing strains metabolized the added xylose completely. Interestingly, the strain *C. beijerinckii* [pMTL83251_P*_thlA_*_1,3-PDO.diolis] consumed only 11 mM xylose (Figure 6D). 1,3-PDO was the main product in case of the induced *C. beijerinckii* [pMTL83251_P*_bgaL_*_1,3-PDO.diolis] strain and *C. beijerinckii* [pMTL83251_P*_thlA_*_1,3-PDO.diolis] (Figure 6E). In contrast to the approaches before, the by-product concentrations varied. Matching the low xylose consumption, *C. beijerinckii* [pMTL83251_P*_thlA_*_1,3-PDO.diolis] produced a low amount of butyrate (30 mM). However, more acetate (10.9 mM) was obtained than by most of the other tested strains. The highest acetate concentration was observed for the induced *C. beijerinckii* [pMTL83251_P*_bgaL_*_1,3-PDO.diolis] strain at 18.5 mM. The lowest acetate production was observed for the wild-type strain at only 1.6 mM. The other tested strains formed between 3.9 mM and 2.6 mM acetate. In contrast to the low acetate production, the wild-type strain showed the highest butyrate concentration (54.5 mM). In the culture broth of the induced *C. beijerinckii* [pMTL83251_P*_bgaL_*_1,3-PDO.diolis] strain, only 28.5 mM butyrate was measured. In contrast, the uninduced strain produced 35.8 mM butyrate. *C. beijerinckii* [pMTL83251_P*_pta-ack_*_1,3-PDO.diolis] and *C. beijerinckii* [pMTL83251_P*_bld_*_1,3-PDO.diolis] produced similar amounts of butyrate (48.6 mM and 48.3 mM, respectively). Traces of ethanol were detected in all studied strains (0.4–0.2 mM). The wild-type strain, *C. beijerinckii* [pMTL83251_P*_pta-ack_*_1,3-PDO.diolis], and the uninduced *C. beijerinckii* [pMTL83251_P*_bgaL_*_1,3-PDO.diolis] strain synthesised traces of butanol (0.3–0.1 mM). 1,3-PDO production was increased with all tested promoters compared to the originally used *pta-ack* promoter.

Table 3 shows a summary of the performed batch growth experiments. A distinct increase in 1,3-PDO production could be obtained by using a buffered medium, resulting in an overall improvement of 82%. Furthermore, the use of buffered medium also resulted in a higher yield of 0.71 mol_1,3.PDO_/mol_glycerol_. For those calculations, the mean of all experiments performed in buffered glycerol medium was used. The second improvement could be observed by exchanging the *pta-ack* promoter. The highest yield of 0.79 and 50.4 mM 1,3-PDO was obtained with the strain *C. beijerinckii* [pMTL83251_P*_thlA_*_1,3-PDO.diolis]. Comparing the 1,3-PDO production utilizing unbuffered medium and *C. beijerinckii* [pMTL83251_P*_pta-ack_*_1,3-PDO.diolis], the 1,3-PDO production could be improved by 167% by using the strain *C. beijerinckii* [pMTL83251_P*_thlA_*_1,3-PDO.diolis] in MOPS-buffered glycerol medium.

## 4. Discussion

*C. beijerinckii* DSM 6423 (=*C. beijerinckii* NRRL B-593) is described as a natural 1,3-PDO producer [22,37,45]. However, the strain obtained from the DSMZ was unable to do so due to the loss of the 1,3-PDO dehydrogenase gene. In contrast to the published literature, the *C. beijerinckii* DSM 6423 wild-type strain only produced traces of 1,3-PDO. To reconstitute the 1,3-PDO production of the wild-type strain, a newly published transformation protocol of *C. beijerinckii* was employed to establish recombinant 1,3-PDO production in *C. beijerinckii* DSM 6423. Therefore, the heterologous gene expression of two different 1,3-PDO gene clusters, the effect of buffered growth media on the recombinant strains, and the use of different promoters for the gene expression were examined. The data presented clearly show that higher concentrations of 1,3-PDO could only be produced with recombinant *C. beijerinckii* DSM 6423 strains. In the published study of Gungormusler et al. the wild-type produced up to 131 mM 1,3-PDO (10 g/L) [37]. Genome analysis of *C. beijerinckii* DSM 6423 showed that the 1,3-PDO gene cluster is not complete. The *dhaB1* and *dhaB2* genes encoding a supposedly coenzyme B_12_-independent glycerol dehydratase could be identified. However, a *dhaT* gene encoding a 1,3-PDO dehydrogenase could not be found in the genome of *C. beijerinckii* DSM 6423. It is possible that another gene with a similar function can also convert 3-HPA to 1,3-PDO, resulting in low 1,3-PDO production. For example, alcohol dehydrogenases could also take over the same function due to their broad substrate specificity [46].

Higher concentrations of 1,3-PDO were only achieved using recombinant *C. beijerinckii* strains carrying the 1,3-PDO gene cluster of *C. beijerinckii* DSM 15410. The plasmid-based expression of the 1,3-PDO genes (*dhaBCE* and *dhaT*) of *C. pasteurianum* did not lead to an increase in 1,3-PDO production compared to the wild-type strain and *C. beijerinckii* [pMTL83251]. As mentioned before, the glycerol dehydratase complex of *C. pasteurianum* is coenzyme B_12_-dependent [27]. However, very little is known about the synthesis of coenzyme B_12_ in *C. beijerinckii* DSM 6423. In many cases, supplementation of vitamin B_12_ was necessary to produce 1,3-PDO when a coenzyme B_12_-dependent glycerol dehydratase was used [47]. In this study, even with supplementation of vitamin B_12_ (2.5 mg/L), *C. beijerinckii* [pMTL83251_P*_thlA_*_dhaBCET.Cpas] was unable to produce 1,3-PDO. The reason could be that *C. beijerinckii* DSM 6423 does not harbour the necessary genes for the conversion of vitamin B_12_ to coenzyme B_12_. Furthermore, Fokum et al. recently showed that the addition of vitamin B_12_ in high concentrations (7.5–10 mg/L) to the culture medium has a negative effect on the 1,3-PDO production of *C. beijerinckii* CCIC 22954 [48]. For industrial approaches, it is also more desirable to use a coenzyme B_12_-independent production strain, in order to keep cultivation costs low due to expensive vitamin B_12_ supplementation.

A further increase in 1,3-PDO production was accomplished by buffering the glycerol medium and preventing a fast pH decrease. The activities of enzymes as well as cofactors within the cell depend strongly on the pH. Therefore, a more stable pH throughout the bacterial cultivation is desired [49]. A positive effect of a controlled pH throughout the fermentation of glycerol was already described before [37]. Experiments with mixed cultures showed that the highest 1,3-PDO production yields could be measured at pH 7 and 8 [50]. Furthermore, most studies on the production of 1,3-PDO from glycerol are performed in fermenters with a controlled pH. Thus, it would be interesting to examine our recombinant production strains in a fermentation experiment with a steady pH.

The biggest improvements were made by exchanging the *pta-ack* promoter. Unfortunately, the direct activity test as described before for *C. saccharoperbutylacetonicum* and *Eubacterium limosum* [35,36] could not be applied for *C. beijerinckii* DSM 6423. In preliminary tests, *C. beijerinckii* carrying pMTL83251_P*_bgaL_*_FAST did not show differences in fluorescence compared to the wild-type strain. Thus, growth experiments were performed to examine the effects of different promoters. Although the lactose-inducible *bgaL* promoter of *C. perfringens* did not lead to the highest 1,3-PDO production, the inducible promoter system is almost tight in *C. beijerinckii* DSM6423 as indicated by the low 1,3-PDO concentration of uninduced cultures. Based on the presented results, the *bgaL* promoter can be described as the second inducible promoter for gene expression in *C. beijerinckii* DSM 6423. Another inducible system is based on the *xylB* promoter from *Clostridium difficile* strain 630, in which gene expression is induced by the addition of xylose [31,51]. The *xlyB* promoter was not examined in this study, since xylose was used as substrate in the growth experiments. Due to the close relationship of *C. saccharoperbutylacetonicum* and *C. beijerinckii*, both strains belong to the second clade of solvent-forming clostridia [52], which is why the *bld* promoter was also tested. Furthermore, the *bld* promoter showed the highest activity in *C. saccharoperbutylacetonicum* compared to promoters *thlA*, *bgaL*, and *pta-ack* [36]. Thus, we assumed high activity of the *bld* promoter in *C. beijerinckii*. However, the highest 1,3-PDO concentration and yield were detected in the case of *C. beijerinckii* [pMTL83251_P*_thlA_*_1,3-PDO.diolis] harboring the *thlA* promoter. This promoter also showed high activity in *C. saccharoperbutylacetonicum* [36].

The use of recombinant clostridial strains to produce 1,3-PDO was already reported before. In 2005, Gonzalez-Pajuelo et al. published an article using genetically modified *C. acetobutylicum* strains [53]. *C. acetobutylicum* was chosen as a host strain, as no genetic tools were available for *C. butyricum* at that time. The recombinant *C. acetobutylicum* strain was engineered for the heterologous production of 1,3-PDO using the genes from *C. butyricum*. Since the described fermentation approaches were fed-batch experiments, a comparison to this study is difficult. The modified *C. acetobutylicum* strain produced more 1,3-PDO compared to the tested strains of the present study due to the different cultivation method. However, all the recombinant *C. beijerinckii* strains presented here showed higher yields when grown in buffered glycerol medium. In comparison with other wild-type clostridial strains, the obtained concentrations are lower [7], but the yield is higher in the case of *C. beijerinckii* [pMTL83251_P*_thlA_*_1,3-PDO.diolis]. As mentioned before, a comparison of batch growth cultures and fermentation experiments under controlled conditions (pH) are hardly possible. However, continuous culture experiments with the newly constructed strains might even show higher 1,3-PDO production, thus proving the way for a cost-competitive bioprocess, when compared to chemical synthesis routes.

## Figures and Tables

**Figure 1 microorganisms-11-00784-f001:**
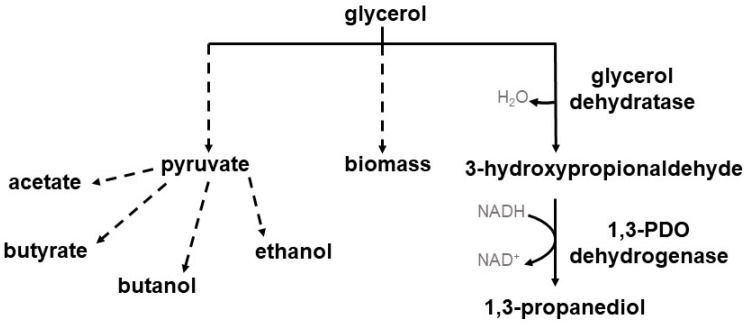
Overview of the glycerol consumption pathway in clostridial species. Glycerol can either be converted reductively, forming 3-HPA and finally 1,3-PDO, or used for biomass and the production of byproducts.

**Figure 2 microorganisms-11-00784-f002:**
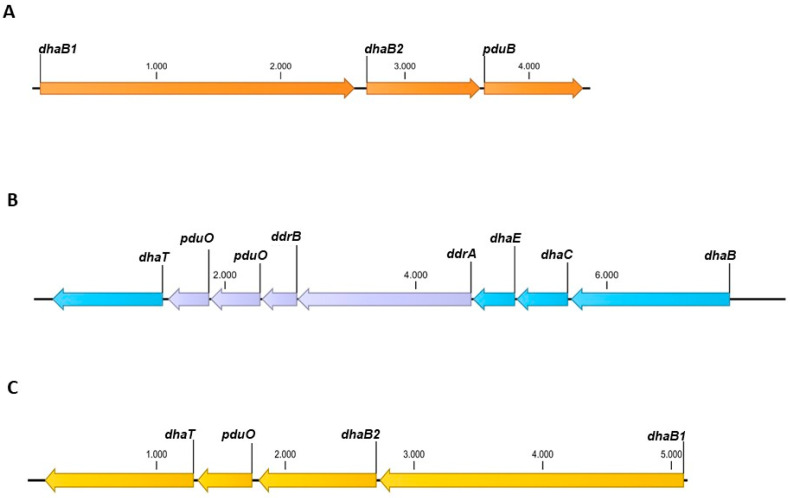
Overview of the 1,3-PDO gene clusters from *C. beijerinckii* DSM 6423 (**A**), *C. pasteurianum* DSM 525 (**B**) and *C. beijerinckii* DSM 15410 (**C**). Genes in blue and yellow were incorporated into the plasmids pMTL83251_P*_thlA_*_dhaBECT.Cpas and pMTL83251_P*_pta-ack_*_1,3-PDO.diolis, respectively. *dhaB1/B2*, genes for coenzyme B_12_-independent glycerol dehydratase; *pduB*, gene for propanediol utilization microcompartment protein; *dhaBCE*, genes for coenzyme B_12_-dependent glycerol dehydratase; *ddrA*, gene for glycerol reactivation factor large subunit; *ddrB*, gene for glycerol reactivation factor small subunit; *pduO*, gene for ATP:cob(I)alamin adenosyltransferase; *dhaT*, gene for 1,3-PDO dehydrogenase.

**Figure 3 microorganisms-11-00784-f003:**
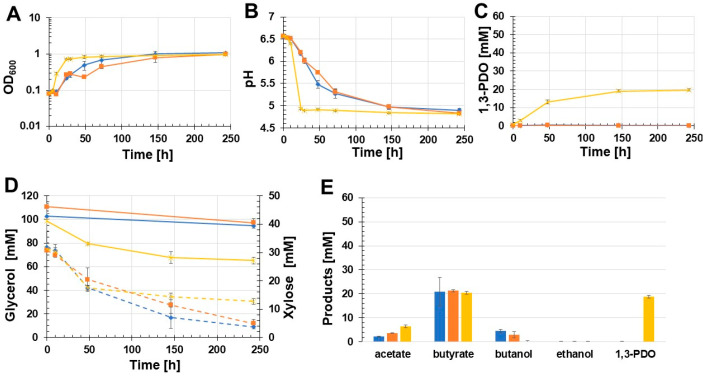
Results of the batch growth experiment with unbuffered glycerol medium performed with the recombinant strains *C. beijerinckii* [pMTL83251_P*_pta-ack_*_1,3-PDO.diolis], *C. beijerinckii* [pMTL83251], and the wild-type *C. beijerinckii* strain. OD_600_ (**A**), pH (**B**), 1,3-PDO production (**C**), substrate consumption (**D**), and concentration of products (**E**). Wild-type: blue rhombus; *C. beijerinckii* [pMTL83251]: orange squares; *C. beijerinckii* [pMTL83251_P*_pta-ack_*_1,3-PDO.diolis]: yellow crosses; glycerol consumption: solid lines; xylose consumption: dashed lines; error bars represent standard deviations, *n* = 3.

**Figure 4 microorganisms-11-00784-f004:**
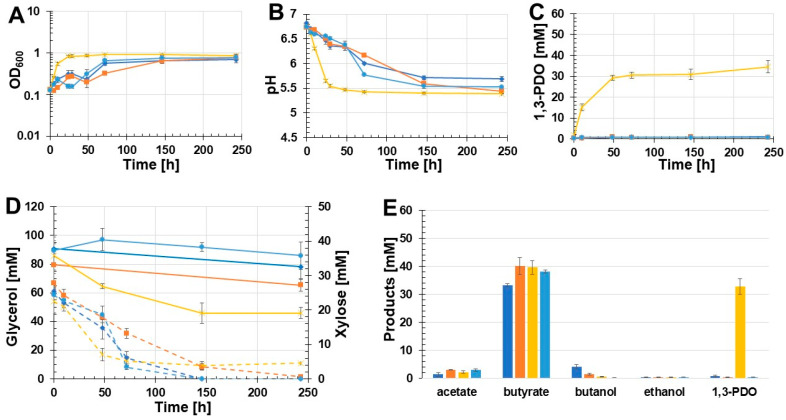
Results of the batch growth experiment using MOPS buffered glycerol medium performed with the recombinant strains *C. beijerinckii* [pMTL83251_P*_thlA_*_dhaBCET.Cpas], *C. beijerinckii* [pMTL83251_P*_pta-ack_*_1,3-PDO.diolis], *C. beijerinckii* [pMTL83251], and the wild-type *C. beijerinckii* strain. OD_600_ (**A**), pH (**B**), 1,3-PDO production (**C**), substrate consumption (**D**), and concentration of products (**E**). Wild-type: blue rhombus; *C. beijerinckii* [pMTL83251]: orange squares; *C. beijerinckii* [pMTL83251_P*_pta-ack_*_1,3-PDO.diolis]: yellow crosses; *C. beijerinckii* [pMTL83251_P*_thlA_*_dhaBCET.Cpas]: light blue circles; glycerol consumption: solid lines; xylose consumption: dashed lines; error bars represent standard deviations; *n* = 3.

**Figure 5 microorganisms-11-00784-f005:**
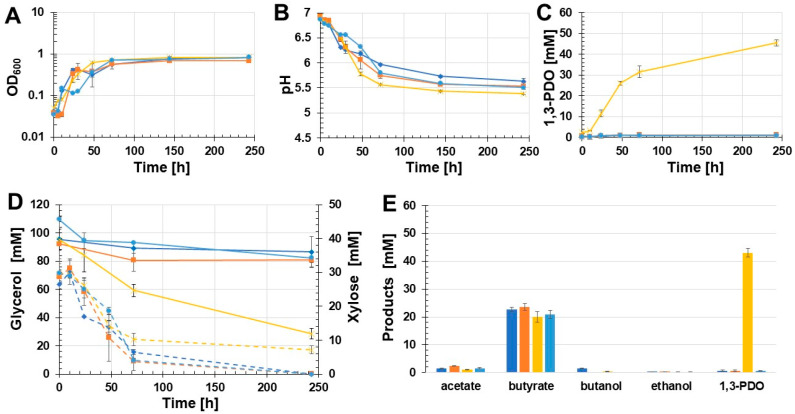
Results of the batch growth experiment using MOPS buffered glycerol medium performed with the recombinant strains *C. beijerinckii* [pMTL83251_P*_thlA_*_dhaBCET.Cpas], *C. beijerinckii* [pMTL83251_P*_pta-ack_*_1,3-PDO.diolis], *C. beijerinckii* [pMTL83251], and the wild-type *C. beijerinckii* strain. The medium of *C. beijerinckii* [pMTL83251_P*_thlA_*_dhaBCET.Cpas] was supplemented with vitamin B_12_. OD_600_ (**A**), pH (**B**), 1,3-PDO production (**C**), substrate consumption (**D**), and concentration of products (**E**). Wild-type: blue rhombus; *C. beijerinckii* [pMTL83251]: orange squares; *C. beijerinckii* [pMTL83251_P*_pta-ack_*_1,3-PDO.diolis]: yellow crosses; *C. beijerinckii* [pMTL83251_P*_thlA_*_dhaBCET.Cpas]: light blue circles; glycerol consumption: solid lines; xylose consumption: dashed lines; error bars represent standard deviations; *n* = 3.

**Figure 6 microorganisms-11-00784-f006:**
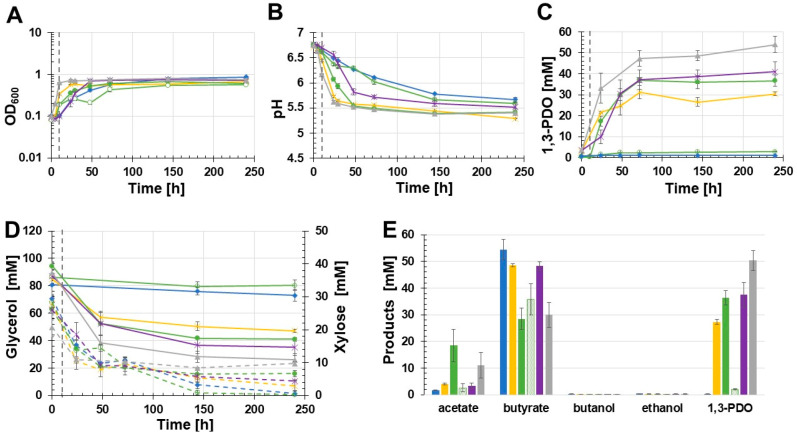
Results of the batch growth experiment using MOPS buffered glycerol medium performed with the recombinant strains *C. beijerinckii* [pMTL83251_P*_pta-ack_*_1,3-PDO.diolis], *C. beijerinckii* [pMTL83251_P*_bgaL_*_1,3-PDO.diolis], *C. beijerinckii* [pMTL83251_P*_bld_*_1,3-PDO.diolis], *C. beijerinckii* [pMTL83251_P*_thlA_*_1,3-PDO.diolis], and the wild-type *C. beijerinckii* strain. OD_600_ (**A**), pH (**B**), 1,3-PDO production (**C**), substrate consumption (**D**), and concentration of products (**E**). Wild-type: blue rhombus; *C. beijerinckii* [pMTL83251_P*_pta-ack_*_1,3-PDO.diolis]: yellow crosses; *C. beijerinckii* [pMTL83251_P*_bgaL_*_1,3-PDO.diolis] induced: green circles; *C. beijerinckii* [pMTL83251_P*_bgaL_*_1,3-PDO.diolis] uninduced: green empty circles; *C. beijerinckii* [pMTL83251_P*_bld_*_1,3-PDO.diolis]: purple stars; *C. beijerinckii* [pMTL83251_P*_thlA_*_1,3-PDO.diolis]: gray triangle; glycerol consumption: solid lines; xylose consumption: dashed lines; induction of gene expression using 20 mM lactose: gray vertical dash lines; error bars represent standard deviations, *n* = 3.

**Table 1 microorganisms-11-00784-t001:** Bacterial strains and plasmids used in this study.

Strain or Plasmid	Features	Reference
*C. beijerinckii* DSM 6423	wild-type	DSMZ * GmbH, Brunswick, Germany
*E. coli* XL1-Blue MRF’	Δ(*mrcA*)183, Δ(*mrcCB-hsdSMR-mrr*)173, *endA1*, *supE44*, *thi-1*, *recA1*, *gyrA96*, *relA1*, *lac*, [F’ *proAB lacI*^q^ ZΔ*M15* Tn10 (Tet^R^)]	Agilent Technologies, Santa Clara, CA, USA
*E. coli* SCS110	*rpsL*, (Str^R^), *thr*, *leu*, *endA*, *thi-1*, *lacy*, *galK*, *galT*, *ara*, *tonA*, *tsx*, *dam*, *dcm*, *supE44D*, (*lac-proAB*), [F’ *traD36 proAB lacI*^q^ZΔ*M15*]	Agilent Technologies, Santa Clara, CA, USA
pMTL83251	clostridial shuttle vector, Em^R^	[34]
pMTL83251_P*_thlA_*_FAST	pMTL83251 containing P*_thlA_* from *C. acetobutylicum* and *feg* (FAST-encoding gene)	[35]
pMTL83251_P*_bgaL_*_FAST	pMTL83251 containing P*_bgaL_* from *C. perfringens* and *feg*	[35]
pMTL83251_P*_bld_*_FAST	pMTL83251 containing P*_bld_* from *C. saccharoperbutylacetonicum* and *feg*	[36]
pMTL83251_P*_pta-ack_*_1,3-PDO.CLOBI	pMTL83251 containing 1,3-PDO gene cluster from *C. beijerinckii* DSM 6423 and *pta-ack* promoter from *C. ljungdahlii*	This study
pMTL83251_P*_pta-ack_*_1,3-PDO.diolis	pMTL83251 containing 1,3-PDO gene cluster from *C. beijerinckii* DSM 15410 and *pta-ack* promoter from *C. ljungdahlii*	This study
pMTL83251_P*_thlA_*_dhaBCET.Cpas	pMTL83251 containing *dhaBCE* and *dhaT* from *C. pasteurianum* DSM 525 and *thlA* promoter from *C. acetobutylicum*	This study
pMTL83251_P*_bgaL_*_1,3-PDO.diolis	pMTL83251 containing 1,3-PDO gene cluster from *C. beijerinckii* DSM 15410 and *bgaL* promoter from *C. perfringens*	This study
pMTL83251_P*_thlA_*_1,3-PDO.diolis	pMTL83251 containing 1,3-PDO gene cluster from *C. beijerinckii* DSM 15410 and *thlA* promoter from *C. acetobutylicum*	This study
pMTL83251_P*_bld_*_1,3-PDO.diolis	pMTL83251 containing 1,3-PDO gene cluster from *C. beijerinckii* DSM 15410 and *bld* promoter from *C. saccharoperbutylacetonicum*	This study

* German Collection of Microorganisms and Cell Cultures (DSMZ).

**Table 2 microorganisms-11-00784-t002:** Primers used in this study.

Primer	Sequence	Purpose
dhaB1/2CoT.diol_fwd	ttaaatttaaagggaggactctagaatgataagtaaaggatttagtacc	amplification of 1,3-PDO gene cluster from *C. beijerinckii* DSM 15410
dhaB1/2CoT.diol_rev	gcaggcttcttatttttatgctagcttaataagcagctttaaatatatttacg	
PthlA.Cpas_fwd	cgaattcgagctcggtacccgggtcaagaagaggcacctcatc	amplification of *thlA* promoter from *C. acetobutylicum*
PthlA.Cpas_rev	ccctcctggtcaccaaattttgatacggggtaacag	
dhaBCE_fwd	aaatttggtgaccaggaggggatcccatgaagtcaaaacgatttcaag	amplification of *dhaBCE* from *C. pasteurianum* DSM 525
dhaBCE_rev	aattcctcctctagtcctctattctaactttatttc	
dhaT_fwd	agaggactagaggaggaattataaaatgagaatg	amplification of *dhaT* gene from *C. pasteurianum* DSM 525
dhaT_rev	atggacgcgtgacgtcgactttaaaatgcttctctaaatattttaactatatc	
PbgaL_1,3-PDO_fwd	cgaattcgagctcggtacccgggtaatttagatattaattctaaattaagtgaaat	amplification of *bgaL* promoter from pMTL83251_P*_bgaL_*_FAST
PbgaL_1,3-PDO_rev	taaatcctttacttatcattctcgagaccctcccaatacatttaaaataa	
PthlA_1,3-PDO_fwd	cgaattcgagctcggtacccgggtcaagaagaggcacctcatc	amplification of *thlA* promoter from pMTL83251_P*_thlA_*_FAST
PthlA_1,3-PDO_rev	taaatcctttacttatcattctcgagacctcctaaattttgatacgg	
Pbld_1,3-PDO_fwd	cgaattcgagctcggtacccggggatatttcccccataagtaaag	amplification of *bld* promoter from pMTL83251_P*_bld_*_FAST
Pbld_1,3-PDO_rev	taaatcctttacttatcattctcgagtcctccttatgatttaaaaattaataac	
16S-27F	ataagcttggatccagagtttgatcctggctcag	amplification of 16S rRNA gene
1492r	actcgaggatatcggttaccttgttacgactt	

**Table 3 microorganisms-11-00784-t003:** Summary of all performed batch growth experiments.

1,3-PDO Producing *C. beijerinckii* Strain	Glycerol Consumption [mM]	1,3-PDO Production [mM]	Yield [mol_1,3-PDO_/mol_glycerol_	Improvement [%] ^1^
*C. beijerinckii* [pMTL83251_P*_pta-ack_*_1,3-PDO.diolis] (unbuffered glycerol medium)	33.3	18.9	0.57	---
*C. beijerinckii* [pMTL83251_P*_pta-ack_*_1,3-PDO.diolis] (MOPS buffered glycerol medium)	48.3 ^2^	34.4 ^2^	0.71 ^2^	82 ^2^
*C. beijerinckii* [pMTL83251_P*_bgaL_*_1,3-PDO.diolis] induced	53.1	36.3	0.68	92
*C. beijerinckii* [pMTL83251_P*_bld_*_1,3-PDO.diolis]	52.1	37.4	0.72	98
*C. beijerinckii* [pMTL83251_P*_thlA_*_1,3-PDO.diolis]	63.4	50.4	0.79	167

^1^ Compared to 1,3-PDO production in unbuffered glycerol medium. ^2^ Average values of all performed batch growth experiments with buffered glycerol medium.

## Data Availability

The data presented in this study are available on request from the corresponding author.

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
