# Peer review of "Heterologous 1,3-Propanediol Production Using Different Recombinant Clostridium beijerinckii DSM 6423 Strains"

_microorganisms, 2023, doi:10.3390/microorganisms11030784_

Round 1
Reviewer 1 Report
The manuscript microorganisms-2259052 is a well-structured paper dealing with Heterologous production of 1,3-propanediol using several recombinant Clostridium beijerinckii DSM 6423 strains. In particular, genes from Clostridium pasteurianum DSM 525 and Clostridium beijerinckii DSM 15410 were introduced into C. beijerinckii DSM 6423 to enable the 1,3-propanediol production from glycerol.
The manuscript is well written, results and conlcusions are clearly presented. Therefore, the manuscript is suitable for publication after minor revisions:
-Fig. 4 D: light ble circles, how do the authors explain the increase in glycerol concentration after the first 50 hours?
Reviewer 2 Report
The improved production of PDO by C. beijerinckii is described in this ms. The increase in production compared to the wt is good. Overall, this is a sound piece of research on chemicals from renewable substrates.
The review will comment mainly on improvement of the presentation, with a couple of technical questions/suggestions that the authors don't nee to address.
Please format the ms. so that tables are not split between two pages - each table should be on just one, the same, page.
lines 15-16. the sentence starting with "Thus, 1,3-PDO...." can be omitted.
Also omit "Therefore" at the beginning of the following sentence.
lines 18 (two places), 194, 270 and 441. Omit the excess "the"s.
line 19 "production by recombinant C. ....".
line 32 Omit the excess "of".
line 41 Omit "other".
line 43 Omit "by".
line 47 "glycerol is generated.... by=product from biodiesel production.".
lines 63, 66, 220, 228 and elsewhere. The "12" should be in subscript.
line 101. Advise that one use free base cys rather than the HCl salt in the future. This was not readily available 20 yr ago but is now.
lines 105-106. Why was MOPS used as the buffer? I've found TES to be the better buffer at neutral pH for growth and PIPES to be better for enzyme assays at neutral pH. Just wondering if there was a reason.
line 107. "E. coli".
lines 182 and 567. separate sentences; e.g., "DNA. After growth....".
lines 188-189. "N2:CO2 (80:20)" better May omit the source of the gas tank.
lines 189-190. Omit the sentence starting with "Afterword".
lines 192-193. "OD600 and pH.... monitored during growth." Better to start the sentence with the subject of the verb rather than with a phrase.
line 193. Omit "Furthermore".
line 198. "18,000 g".
line 203. separate sentences. "HCl. 1 µl....".
lines 206-207. Omit the quotation marks.
line 208. "20 µl...." to maintain consistency of presentation.
line 219 separate sentences. "strains. One is....".
lines 224 and 495. "to reconstitute the....".
line 226. "transformed into....".
line 265. "strains produced....".
line 270. Omit "strain".
line 276. "was only detected in....".
line 281. Omit "growth".
line 283. "Xylose was not completely consumed by any of the strains tested. The main....".
line 288 "produced the most....".
line 292. "However, this strain only produced traces of....".
line 318. "5.5" and "5.4". This shows that the actual pH drop was not drastic.
There are further suggestions to improve the presentation but the authors may make those revisions as needed based on comments to tis point.
Reviewer 3 Report
The manuscript explores the development of a recombinant Clostridium beijerinckii strain for the production of 1,3-propanediol from glycerol, aiming its application in biorefineries. The manuscript is well written and well organized and the ways and means are well described. Some points need attention before being published:
· Please perform a careful revision to avoid typo mistakes.
· Strains and cultivation: please add reference for media compositions
· Analytical methods: please add references to the methods used
· There are some methodologies described in the results section, please reallocate it in methodology section
· Figures 3C and 4C: Y axis maximum value can be fixed at 30 and 40, respectively.
· Lines 305-348 and lines 405-461: these paragraphs are too long, please separate them.
· Figure 5C: why there is no experimental point between 75 and 250h?
· The statistical analysis of results from figure 6 and table 3 are necessary to enrich the discussion and ensure that some conditions are superior than other.
